# Innovative AI-Enhanced Ice Detection System Using Graphene-Based Sensors for Enhanced Aviation Safety and Efficiency

**DOI:** 10.3390/nano14131135

**Published:** 2024-07-01

**Authors:** Dario Farina, Hatim Machrafi, Patrick Queeckers, Patrice D. Dongo, Carlo Saverio Iorio

**Affiliations:** 1Centre for Research and Engineering in Space Technologies (CREST), Department of Aero-Thermo-Mechanics, Université Libre de Bruxelles, 1050 Bruxelles, Belgium; 2GIGA-In Silico Medicine, Université de Liège, 4000 Liège, Belgium

**Keywords:** aerospace icing prevention, risk mitigation in aviation, graphene-based sensors, dynamic ice sensing, deicing, conductive polymer applications, smart sensors, PEDOT:PSS polymers, 2D materials, micromachines

## Abstract

Ice formation on aircraft surfaces poses significant safety risks, and current detection systems often struggle to provide accurate, real-time predictions. This paper presents the development and comprehensive evaluation of a smart ice control system using a suite of machine learning models. The system utilizes various sensors to detect temperature anomalies and signal potential ice formation. We trained and tested supervised learning models (Logistic Regression, Support Vector Machine, and Random Forest), unsupervised learning models (K-Means Clustering), and neural networks (Multilayer Perceptron) to predict and identify ice formation patterns. The experimental results demonstrate that our smart system, driven by machine learning, accurately predicts ice formation in real time, optimizes deicing processes, and enhances safety while reducing power consumption. This solution holds the potential for improving ice detection accuracy in aviation and other critical industries requiring robust predictive maintenance.

## 1. Introduction

Ice formation on aircraft surfaces represents a critical safety hazard, thus significantly impairing aerodynamic performance by increasing drag and reducing lift. This phenomenon, known as icing, occurs when supercooled water droplets in the atmosphere freeze in contact with an aircraft during flight. The presence of ice can lead to increased fuel consumption, reduced engine performance, and even catastrophic failure of aircraft control systems. Consequently, the development of effective deicing and anti-icing technologies is crucial to aviation safety and operational efficiency [1,2].

The primary technologies currently in use include pneumatic systems, electrothermal heating elements, fluid-based anti-icing, and hybrid ice protection technologies [3,4,5].

Pneumatic systems work by inflating rubber boots on leading edges to break the ice. They can be manually or automatically activated, and their effectiveness relies on timely activation to prevent ice buildup [6]. Electrothermal heating elements, embedded in critical areas of the aircraft surface, melt and remove ice by utilizing electrical heating elements. While efficient, they consume a significant amount of power and often require continuous activation [7].

Fluid-based anti-icing involves spraying glycol-based solutions onto aircraft surfaces, thus preventing ice formation by lowering the freezing point. However, their efficacy is limited to specific flight conditions, and they require regular reapplication [8]. Hybrid ice protection technologies integrate pneumatic, electrothermal, and fluid-based methods to create a comprehensive solution. Although these systems are effective due to the combination of techniques, their complexity and energy demands can be prohibitive [9].

Despite advancements in pneumatic, electrothermal, fluid-based, and hybrid anti-icing systems, these solutions are often limited by manual activation, high energy consumption, or complexity. As a result, there is a growing demand for automated, predictive, and energy-efficient ice detection and removal systems. Our research aims to address this gap by developing an innovative smart ice control system using machine learning models to predict ice formation. This system should work as an interpreter for the signal coming from a graphene-based sensor [10,11,12], as well as a provider of local heating only when necessary.

Graphene, a two-dimensional material with exceptional properties, offers significant potential in developing advanced ice detection systems. Graphene’s high thermal and electrical conductivity, combined with its mechanical strength and flexibility, make it an ideal material for sensor applications [13,14]. In particular, graphene can efficiently conduct heat and electricity, which is crucial for real-time ice detection and actuation systems. These unique properties of graphene have been extensively reviewed in the context of sensor applications [15,16,17,18].

In this study, we focus on utilizing graphene-based thermoresistors for ice detection in aircraft. Our system leverages the unique properties of graphene to enhance the accuracy and efficiency of ice detection. The system integrates machine learning models to predict ice formation patterns, thereby optimizing deicing processes and reducing power consumption. This approach not only improves safety but also contributes to the operational efficiency of aircraft.

The characterization of graphene is essential to validate its suitability for this application. Instead of performing our own characterization, we utilized graphene that was previously characterized and detailed in the work by Sibilia et al. (2023). In their study, the graphene nanoplatelets (GNPs) were produced through thermal expansion and liquid exfoliation, followed by a thorough analysis using various techniques. Raman spectroscopy confirmed the quality and number of graphene layers, while Scanning Electron Microscopy (SEM) was employed to examine the surface structure of the graphene sheets. Additionally, the electrical and thermal conductivities were assessed, thus validating the high quality and suitability of the graphene used in our sensors [19].

Furthermore, the functionalization of graphene was carried out to enhance its performance. Chemical functionalization involves introducing functional groups to improve dispersion in the composite material. Physical treatments, such as plasma treatment, were applied to modify the surface properties of graphene, thus enhancing its interaction with the matrix material and overall sensor performance [20,21,22,23].

Supercooling is crucial to the method of detecting ice formation because the freezing process is exothermic and releases heat, thus causing a temperature rise in the surrounding area. Substantial heat is released in a short time during a freezing event on the airplane wing surface, and this thermal anomaly can be detected using thermocouples without prior knowledge of the precise freezing point. The developed device detects the presence of ice and activates melting via the same graphene-based resistance system.

This research is significant for several reasons. First, using machine learning algorithms allows for the accurate detection and prediction of ice formation patterns, thus reducing false positives and enabling timely deicing actions. By leveraging supervised and unsupervised machine learning algorithms, as well as neural networks, the goal is to optimize the deicing process in real time while reducing power consumption. Machine learning techniques such as decision trees, support vector machines, and neural networks are effective for this purpose, thus providing robust models for classification and prediction tasks [24,25,26,27].

The smart ice control system presented in this research lays the foundation for further exploration into predictive ice detection and control. Future developments could include enhanced integration with avionics for seamless communication between the ice control system and other critical flight systems, adaptive learning algorithms that continuously refine the system’s ice detection and removal capabilities based on in-flight data, and the extension of this technology to other industries where ice formation poses a challenge, such as wind turbines and power lines. Ultimately, this research advances predictive ice detection and control, thus contributing to safer and more efficient operations in aviation and beyond.

## 2. Experimental Setup and Methodology

### 2.1. Concept of Investigation and Sensor Preparation

The methodology employed in this study is based on one fundamental principle of thermodynamics [28]. The freezing process of water does not always commence immediately when the temperature (T) drops below its freezing point (T_*f*_) but often begins at a significantly lower temperature, which is a phenomenon known as supercooling [29].

In fact, water freezing is an exothermic process, thereby releasing heat that influences its surroundings. During icing events on aircraft wings, a significant amount of heat is released over a short period. By using thermocouples to measure the temperature of the wing surface, the instant heat release can be captured without prior knowledge of the freezing point. Our device detects ice formation and melts it by sending a current through a graphene resistor.

In the temperature profiles captured via thermocouples, “Temp1” and “Temp2”, indicate ice detection in the so-called “passive method”. The term “passive” is used because no active electrical signal is sent through the sensing area; instead, the system observes variations in terms of resistance and sharp changes in temperature, which allow for the analysis of the events occurring on the surface. These temperature changes are different from those caused by evaporation or external temperature fluctuations. Indeed, temperature changes due to freezing are much more abrupt than by the other aforementioned mechanisms. By monitoring the temperature profiles, the onset and development of ice formation can be detected, thus providing critical data for understanding and mitigating ice accretion on aircraft surfaces. For wet (glaze) ice, the stagnation region remains at 0 °C due to the latent heat released during freezing. For dry (rime) ice, the temperature peaks below freezing. A characteristic temperature profile can be measured using thermocouples isolated from the ice accretion area by prepreg composite materials [30].

Prepregs are composite materials consisting of reinforcement fibers preimpregnated with a thermoplastic or thermoset resin matrix. Cured under high temperature and pressure, they exhibit unique properties such as high strength, uniformity, and lightweight nature. The prepreg used in this project is Fibre Glast Prepreg, which is shipped between layers of backing cloth to prevent sticking. It is partially cured for handling and typically stored in cool conditions.

First, it is necessary to prepare the shape required for the experiment, which is a square of 11 cm. The prepreg is partially cured, which means that it is in the B-stage. It can be trimmed, pleated, and formed using a knife or a pair of scissors. To obtain a glossy finished component, it is necessary to have a flawless mold, which has been waxed and coated with PVA.

Then, it is necessary to peel off the backing cloth from one side and adapt this side to the mold. The prepreg surface is very sticky, so it is important to place the material carefully into the mold. Once the layer is fixed over the mold, it should not be moved anymore.

Two sheets of Teflon have been used as a mold (Teflon was selected as a mold material due to its nonstick properties, high thermal stability, and chemical inertness, which prevent contamination and ensure easy release of the composite material without damaging the graphene-based sensors), and the layers of prepreg have been placed in the middle to ensure a smooth surface. It is possible to continue placing layers on top of each up to a total thickness of 1/4 inch (0.654 cm) at a single time. The prepreg structure used in this work is multilayer; thermocouples and graphene strips have been placed between the layers.

After the first layer is fixed on the mold, the second cloth is peeled off, and another piece of graphene measuring 0.5 cm is placed over the first layer. Each graphene strip used in the experiment measured approximately 0.5 cm in width and varied in length up to 5 cm, depending on the specific test setup. The graphene employed was multilayered, with a thickness ranging from 70 µm to 85 µm. These strips were free-standing and directly integrated into the glass–fiber reinforced polymer (GFRP) without any supporting substrate. The setup prepared is composed of three layers; between the second and the third layer were placed 2 thermocouples.

Once the layers and the thermocouples are placed in the mold, the prepreg must be compressed in order to allow bonding, remove any trapped air between the layers, and squeeze out any excess resin. To do this, the prepreg has been compressed with 2 steel plates. The resin will naturally become thinner as the temperature is raised, and there will be resin flow prior to a full cure of the resin.

Heat and pressure are essential to completely cure the prepreg. While required pressure can be achieved using a press, an autoclave with temperature and pressure controls is ideal. Generally, an oven is needed to manage temperature ramp-up, ramp-down, and maintain uniform temperature throughout the curing process.

All curing cycles start with a temperature ramp-up and conclude with a ramp-down. The variation lies in the target temperature and the duration needed for a complete cure. Temperature is maintained throughout the cycle and then reduced at a rate of less than 5 °C/min to at least 150 °C before the part is removed from the oven.

After curing, the part cools at room temperature and is then removed from the mold. If the prepreg is not sturdy or remains sticky, adjustments to curing time or temperature may be necessary.

Figure 1 illustrates the preparation and setup of the sensor and system. Part (a) shows a clear view of the sensor preparation after the stage of compression, thus detailing the arrangement of cables, thermocouples, graphene resistance, and prepreg layers. Part (b) presents the flow diagram of the system, thus depicting the sequence from temperature sensing to ice detection and the activation of the graphene resistance for temperature control.

Figure 2 illustrates the synchronized temperature profiles from two sensors, “Temp1” and “Temp2”, thus showing the progression of ice formation and melting over time. Key stages, such as supercooled water formation, latent heat release, and icing development, are annotated to highlight crucial temperature changes that indicate the presence of ice. The heating phase, where applied heat melts the formed ice, is evident in the gradual temperature rise. This collective view of the temperature profiles helps demonstrate the effectiveness of ice detection using dual thermocouples and shows how the system accurately identifies and manages icing events through distinct temperature patterns.

### 2.2. Experimental Setup

The central objective of this work is the development of a smart ice detection and control device capable of autonomous operation to minimize operator intervention. This system leverages existing technologies while ensuring simplicity, cost-effectiveness, and a streamlined design. Compatibility between Arduino (an open source electronics platform for building digital devices) and Raspberry Pi (a small single-board computer for various programming tasks) was assessed to identify the optimal configuration for data collection and system management. It was determined that both Arduino uno and Raspberry Pi 3b+ could function complementarily, with Arduino handling data collection (specifically temperature data), while Raspberry Pi manages the system and hosts machine learning models.

Graphene-based materials have shown significant promise for various smart sensor applications, including electronic noses (e-noses) for detecting volatile organic compounds. For example, Rabchinskii et al. (2023) explored the use of graphene in sensors for environmental monitoring, thus highlighting its sensitivity and versatility [31]. Similarly, Tan and Ang (2021) reviewed advancements in graphene-based electronic materials for sensors and highlighted their potential in flexible and wearable devices [32]. These studies demonstrate the broad applicability and effectiveness of graphene-based materials in enhancing sensor performance, which supports the rationale for their use in our ice detection system.

Figure 3 illustrates the scheme of the smart ice detection system. Raspberry Pi was selected for its comprehensive computer-like functionality and versatility, including a powerful microprocessor, large memory, and versatile I/O ports, thereby making it ideal for managing the system and implementing predictive ice detection algorithms. Arduino, serving as a microcontroller, acquires temperature data and communicates these data to Raspberry Pi via a serial port. The thermocouple amplifier that was used offers optimal sensitivity and a broad temperature range, thus enabling effective data collection during ice formation studies.

An eight-channel relay module acts as a switch, thus activating heating elements during passive deicing methods. For experiments requiring more than two thermocouples, a CD74HC4067 multiplexer was incorporated to extend data collection capabilities. An Adafruit MAX31856 was used to amplify the signal coming from the sensing area.

The sensing area was composed of graphene strips GS50-type produced by NANESA SRL (Arezzo, Italy). In fact, graphene is a unique material known as a zero-gap semiconductor, because its conduction and valence bands meet at the Dirac points. This property allows it to conduct electrical charges through both electrons and holes, thereby providing ambipolar conduction. In terms of thermal properties, graphene relies on phonons for heat conduction, as with other semiconductors, and achieves the highest thermal conductivity in its single-layer form. Mechanically, graphene is incredibly strong yet lightweight and flexible, thus making it highly resistant to mechanical stress. Its optical properties include high transparency despite its ability to absorb a fraction of incident light. These electrothermal properties make graphene an innovative, versatile material suitable for applications requiring excellent conductivity and durability. The electrical and mechanical properties of graphene make it ideal for conducting current in the ice mitigation process. Graphene’s high thermal conductivity enables rapid heating to melt detected ice accumulations. Graphene has been embedded between two prepreg sheets, as described in Section 2.1.

Red and green LEDs provide visual indicators of the system state, while an I2C LCD screen displays operational messages. The final configuration integrates all components within a metallic enclosure for organized port access and visual indicators.

## 3. Machine Learning for Ice Detection and Control

Machine learning algorithms can effectively detect and control ice formation by leveraging a combination of Python 3.7 programming tools and neural network architectures. Python, a high-level programming language, offers a rich ecosystem of open source libraries suitable for scientific computing and data analysis. This versatility enables the development of sophisticated machine learning models using libraries like NumPy, Pandas, SciPy, and scikit-learn. In fact, machine learning (ML) algorithms are broadly classified into supervised, unsupervised, semisupervised, and reinforcement learning, each with having unique applications depending on the nature of the problem being solved. Supervised learning relies on labeled data to train predictive models, while unsupervised learning seeks patterns in unlabeled datasets to identify clusters and relationships. Semisupervised learning combines elements of both approaches, and reinforcement learning enables models to improve through a feedback loop of rewards and penalties.

### 3.1. Python Programming and Libraries

Python’s high-level, object-oriented programming facilitates concise and efficient code writing, while its extensive modules and packages provide robust tools for data analysis and visualization. Key libraries used in this project include the following:**SciPy:** An open-source library for scientific computing offering modules for optimization, linear algebra, signal processing, and special functions.**NumPy:** A numerical computing library that adds support for multidimensional arrays and high-level mathematical functions.**Pandas:** A data manipulation library designed for numerical tables and time series.**Matplotlib:** A plotting library that integrates with NumPy to provide an API for embedding plots in applications.**scikit-learn:** A machine learning library featuring algorithms for classification, regression, clustering, and model evaluation.

A supervised machine learning approach using labeled training data is highly effective for ice detection tasks. Various algorithms such as Logistic Regression, Random Forests, Support Vector Machines (SVMs), and Artificial Neural Networks (ANNs) excel in binary classification and regression tasks, thus making them suitable for this application. In our research, we will use and compare these methods to determine the most efficient algorithm for predicting ice formation.

Among these, the K-Means algorithm, which is an unsupervised clustering method, will be used to group data based on their proximity to k randomly selected centroids. It uses distance measurements like Euclidean, Manhattan, Chebyshev, and Minkowski to determine cluster membership.

Logistic Regression, a supervised binary classification model, will be employed to predict discrete outcomes based on independent variables using a probability function. This model is particularly effective in situations where the outcome variable is binary, such as detecting the presence or absence of ice.

Decision Trees, which are hierarchical models used for classification and regression tasks, will be used to organize decision rules into nodes and split data based on criteria such as information gain and the Gini index, thus creating a tree-like structure that is easy to interpret.

Random Forests, an ensemble learning method, will enhance the performance of decision trees by building multiple trees and averaging their predictions. This approach reduces overfitting and improves accuracy.

Support Vector Machines (SVMs), which excel in classification and regression problems by maximizing the geometric margins between classes, will be used for their effectiveness in high-dimensional spaces.

Artificial Neural Networks (ANNs), inspired by biological neurons, will be used for their capability to learn data patterns through interconnected layers. These networks, consisting of input, hidden, and output layers, can be adapted for various tasks, including classification, regression, and clustering.

Advanced neural network architectures will further extend the capabilities of ANNs to tackle more complex tasks. Convolutional Neural Networks (CNNs) will process multidimensional input through convolutional, pooling, and fully connected layers, thus making them highly effective for image classification and object detection. Recurrent Neural Networks (RNNs), which maintain hidden states through time, will be useful for processing sequential data. Long Short-Term Memory (LSTM), an advanced form of RNN, will incorporate memory cells to learn long-term dependencies, enhancing its ability to model temporal sequences [33,34].

In conclusion, our study will employ and compare these machine learning algorithms to determine the most effective approach for detecting and controlling ice formation, thus ensuring aircraft safety and operational efficiency.

### 3.2. Data Collection and Ice Detection Using Machine Learning Algorithms

This section presents the application of machine learning algorithms to detect ice formation based on the experimental data collected. The study focuses on supervised and unsupervised learning approaches to analyze data from both active and passive methods. The collected data, formatted in comma-separated files and graphical images, underwent machine learning processing.

The data were collected according to the experimental setups detailed in Section 2.2. In the method used, real-time temperature measurements were taken to observe graphical patterns indicative of ice formation, such as the notch-type peak points being visible, as well as a camera positioned for validation. “No ice” and “ice” presence were recorded and labeled with “0” = no Ice, “1” = Ice. Table 1 contains a sample of the data collected through the data collection.

The dataset was processed to be ready for the machine learning algorithms and other ice detection algorithms. The first processing technique applied was the conversion of the time string to double. Missing data or anomaly data that were observed were removed from the data set programmatically.

The dataset was transformed by combining every five consecutive temperature data points into a single point. The modal class for the five data points was considered the target class for the combined point.

The ice detection algorithm was evaluated using data collected during ice formation experiments and in real-time scenarios. For collected data, the algorithms were assessed in the Jupyter Notebook environment on a Windows operating system. The evaluation employed machine learning techniques with 5-fold cross-validation, which involves dividing the dataset into training and testing subsets. The dataset was split into a 1:5 ratio for testing and training. Here are key terms used in performance evaluation:**True Positive (TP):** The relevant class correctly detected by the model (ice formation).**True Negative (TN):** The irrelevant class correctly identified (nonice formation).**False Positive (FP):** The irrelevant class mistakenly detected as relevant.**False Negative (FN):** The relevant class mistakenly detected as irrelevant.

### 3.3. Metrics

We introduce some useful definitions based on the key terms for performance evaluation:**Precision:** Measures the portion of correctly detected relevant data points out of all detected relevant points.
(1)$$Precision=\frac{TP}{TP+FP}$$**Recall:** Measures the correctly detected relevant data points among all existing relevant points.
(2)$$Recall=\frac{TP}{TP+FN}$$**F1 Score:** Balances precision and recall.
(3)$$F1\;\mathrm{score}=2*\frac{Precision*Recall}{Precision+Recall}$$**Accuracy:** Indicates the proportion of correctly detected relevant and irrelevant data points among all data.
(4)$$Accuracy=\frac{TP+TN}{TP+FN+FP+FN}$$

### 3.4. Peak Identification for Ice Detection

Ice detection relies on peak identification in the method used. Peak characteristics like maximum and minimum heights, widths, and prominence are considered to minimize false positives. Figure 4 defines the peak properties, while Table 2 summarizes the values obtained.

### 3.5. Unsupervised Learning Approach

Unsupervised learning aims to identify ice detection patterns based on clustering. The K-Means algorithm was used, where the k value was set to 2, thereby representing two clusters: one for ice and one for no ice. Each cluster has a centroid, which is the central point of the data points within that cluster. The algorithm classifies the data into ice and no-ice clusters by computing the Euclidean distance between each data point and the centroids. The Euclidean distance Dist(t,c) between a data point *t* and a centroid *c* is calculated using the following equation:(5)$$Dist\left(t,c\right)=\sqrt{\sum_{i=1}^{5}{\left(t_{i}-c_{i}\right)}^{2}}$$

In this equation, $t_{i}$ represents the individual data points, and $c_{i}$ represents the coordinates of the centroid. The distance Dist(t,c) helps determine the proximity of each data point to the centroids. Based on this distance, each data point is assigned to the nearest centroid, thereby classifying it into either the ice or no-ice cluster.

Table 3 provides a sample of the fitted data used for clustering, and Figure 5 visualizes the k-means classification, thus showing how the data points are grouped based on their proximity to the centroids.

### 3.6. Supervised Learning Approach

The supervised learning approach involves labeling data points to identify those resulting in ice formation. The algorithms used include K-Nearest Neighbors (KNNs), Logistic Regression, Random Forests, Support Vector Machines (SVMs), and Multilayer Perceptrons.

Logistic regression models classify data based on probabilistic thresholds. Table 4 shows the logistic parameters obtained, and Equation (Equation 6) defines the logistic function.
(6)$$\begin{array}{cc} \ln\left(\frac{\pi }{\pi +1}\right) & =-6.59001252-0.0326796t_{1}+0.0278647t_{2}\\ \; & \;+0.10516002t_{3}+0.04234012t_{4}-0.15719197t_{5} \end{array}$$

## 4. Results

This section outlines the experimental and nonexperimental results collected during the project. The experimental results were obtained from the ice formation experiments discussed above, include the graphs and values. The nonexperimental results consist of the performance evaluation of the ice detection models developed in Python using the Jupyter Notebook environment.

Figure 5 shows the results obtained from the K-Means algorithm. The first temperature is plotted against the last temperature, and the centroid computed by the K-Means algorithm is used to classify the data points. The plot shows that the data points indicating ice detection are well separated from the data points indicating no-ice detection. However, there are false detection points indicated by the colors overlapping on each other.

When the system detects the presence of ice, the program gives a message indicating which thermocouple is affected by ice and the corresponding temperature measured. This is illustrated in Figure 6. Part (a) shows the Python output with the detected ice event. Part (b) presents the temperature profile graph with the red and blue lines representing the two different thermocouples, thus highlighting the ice detection point circled in the graph. Part (c) displays the LCD screen showing the ice detection message and the blue LED indicator turned on. Part (d) shows the prepreg setup with a visible formation of ice over the sensing area. The setup consists of three layers, with graphene strips placed between the layers and thermocouples embedded for temperature measurement. The ice formation demonstrates the effectiveness of the graphene-based sensors in detecting ice.

### Performance Evaluation of Ice Detection Algorithm

Figure 7 shows the heatmap representing the performance metric, which includes precision, recall, F1 score, and accuracy obtained from each of the classifiers. From the map, it is possible to see that the Multilayer Perceptron has the highest accuracy. However, it is necessary to focus on the detection of the relevant class, which is given by precision and recall. Therefore, in order to find a balance between precision and recall, the focus is set on the F1 score. The K-Nearest Neighbor and Decision Tree algorithm had the highest F1 scores. Thus, the algorithms that are effective in detecting ice are the K-Nearest Neighbor (KNN) and the Decision Tree algorithm. The Receiver Operating Characteristic (ROC) curve, which is a graphical plot illustrating the diagnostic ability of a binary classifier system as its discrimination threshold is varied, which is shown for all classifier models in Figure 8.

**Figure 6 nanomaterials-14-01135-f006:**
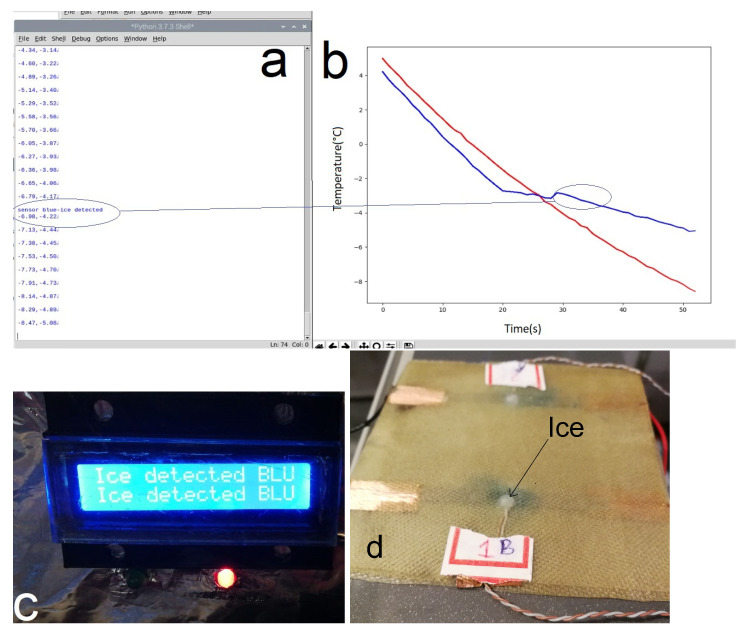
(**a**) Sensor output showing temperature data and ice detection indication in the Python 3.7 shell. (**b**) Temperature profiles with Temp1 (red line) and Temp2 (blue line); the circled area indicates ice formation. (**c**) LCD display and blue LED light confirming ice detection. (**d**) Prepreg setup with visible ice formation over the sensing area illustrating the graphene-based sensors’ ability to detect ice.

## 5. Discussion

The results of this project demonstrate the feasibility and effectiveness of using machine learning algorithms to develop a smart system for detecting and controlling ice formation on aircraft surfaces. Several key findings emerged from the experimental evaluation of the passive and active detection methods, which were tested using supervised and unsupervised learning models.

The unsupervised K-Means Clustering algorithm was able to distinguish between ice and no-ice conditions, but some false positives were noted due to overlapping clusters. This indicates that while K-Means can provide a general classification, the precision could be improved by refining the clustering criteria or supplementing it with additional features.

Supervised learning algorithms, particularly K-Nearest Neighbors (KNNs) and Decision Trees showed the highest precision, thus achieving up to 62% accuracy in predicting ice formation. Labeling of the training data provided better-defined patterns, thus resulting in more accurate predictions. However, further improvements could involve tuning hyperparameters and increasing the volume of labeled training data.

The integration of the Raspberry Pi and Arduino platforms with machine learning algorithms proved advantageous due to their low cost and flexibility. The inclusion of indicator LEDs and an I2C LCD provided valuable real-time feedback to visually confirm the predictions. Despite this success, challenges were noted in managing the active heating process and maintaining consistent temperature measurements across different environmental conditions.

Overall, the results indicate that machine learning models can effectively predict ice formation patterns, and combining these models with a robust hardware platform like Raspberry Pi allows for cost-effective ice detection and control. However, to enhance the precision level of the system, we suggest incorporating additional features that could improve the accuracy and robustness of the ice detection algorithms. One potential feature is the integration of environmental sensors to measure humidity and wind speed, which are critical factors influencing ice formation. Furthermore, employing advanced signal processing techniques to filter out noise and enhance signal clarity could significantly improve the detection accuracy. Another promising approach is to utilize a larger dataset with diverse icing conditions to train the machine learning models, thus ensuring that they can generalize better to various real-world scenarios. Enhancing the dataset with labeled examples of different ice formation patterns will also contribute to the improved precision of the system.

## 6. Conclusions

The primary objective of this project was to develop a low-cost, smart system capable of predicting and mitigating ice formation by using machine learning techniques and artificial neural networks. The project was conducted in three phases: research, implementation, and integration.

In the research phase, various single-board computers, interface boards, and sensors were evaluated for their specifications and suitability. During the implementation phase, sensors, single-board computers, and Python code were integrated and evaluated. Various challenges, such as determining suitable heating elements and active detection systems, were addressed. Ultimately, relay boards were chosen over MOSFETs due to their higher current support and safety.

The integration phase involved designing the smart device, developing algorithms, and interfacing components like the Raspberry Pi and Arduino Mega. Two separate datasets were created for accurate model training. Various machine learning algorithms, including K-Nearest Neighbors and Decision Trees, were developed and evaluated.

The evaluation phase involved training models and validating their accuracy. The system was tested by visually confirming the physical formation of ice on the prepreg and comparing this to predictions displayed on the Raspberry Pi system. Additional features, such as an I2C LCD screen and indicator LEDs, provided real-time feedback. Among the tested models, K-Nearest Neighbors and Decision Trees achieved the highest accuracy, with 60% and 62% precision, respectively.

Implementing graphene-based sensors provided significant advantages for the ice detection system. Graphene’s superior thermal and electrical properties, including high conductivity and large surface area, enhance the sensitivity and accuracy of ice detection. This ensures rapid and localized detection, thus improving the overall responsiveness of the system. The use of graphene not only increases safety by providing timely and precise ice detection but also highlights the innovation in employing advanced nanomaterials for practical aviation safety applications. The innovative aspect of this project lies in the integration of graphene-based sensors with machine-learning algorithms to create a smart ice detection and control system. This system is capable of providing real-time feedback and predictions, thus ensuring enhanced safety and efficiency in aviation operations. Future improvements include expanding the graphene network for localized detection, enhancing the dataset for more accurate model training, and refining the user interface.

## 7. Patents

International patent n° WO2022162068A1/ EP4036565A1 (accessed on 30 May 2024).

## Figures and Tables

**Figure 1 nanomaterials-14-01135-f001:**
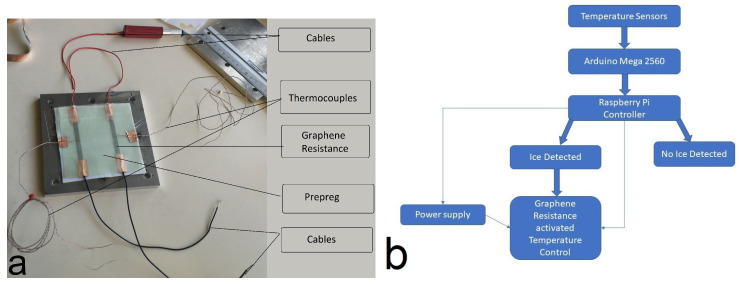
(**a**) Setup of the graphene-based ice detection system. (**b**) Flowchart of the smart ice detection and control system.

**Figure 2 nanomaterials-14-01135-f002:**
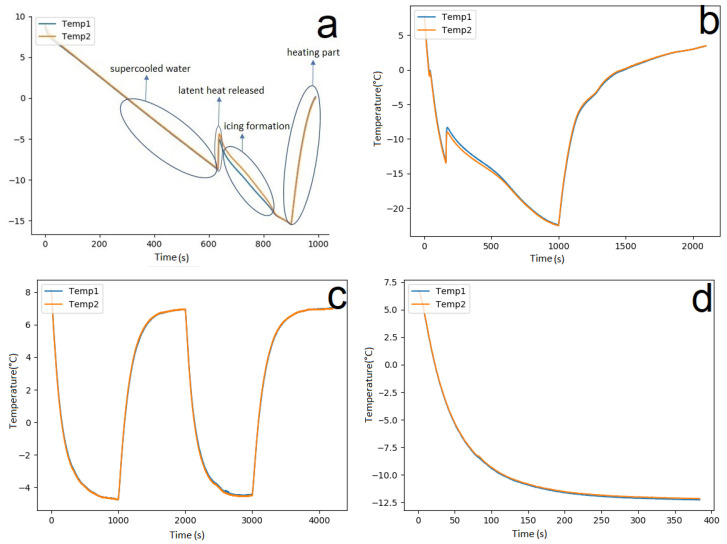
Temperature profiles captured via thermocouples: (**a**) Icing formation with stages like supercooled water and latent heat release annotated. (**b**) Temperature profile indicating ice detection and subsequent melting via resistance activation. (**c**) Cyclic phases have no ice formation. (**d**) Temperature profiles without ice formation.

**Figure 3 nanomaterials-14-01135-f003:**
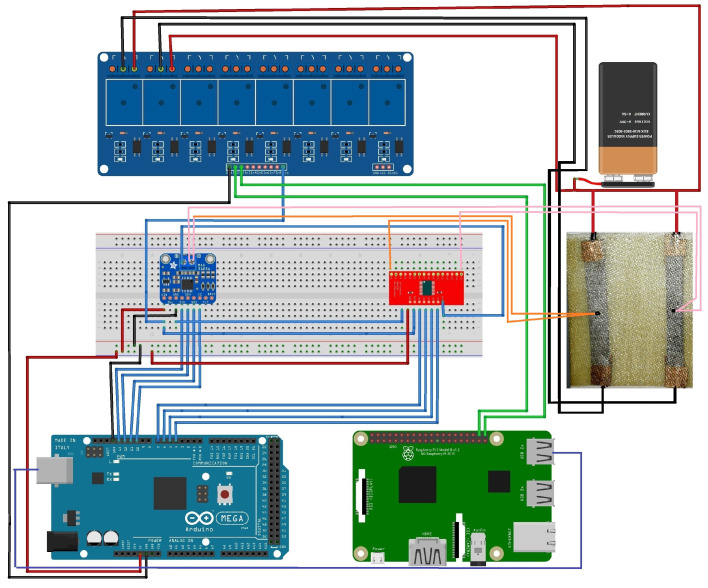
Schematic of the smart ice detection and control system illustrating the connections between the Arduino Mega 2560, Raspberry Pi, relays, temperature sensors, and graphene resistance heating element.

**Figure 4 nanomaterials-14-01135-f004:**
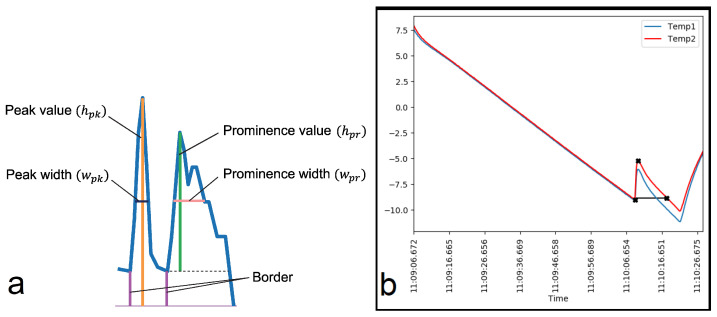
(**a**) Definition of peak properties. (**b**) Ice detection point.

**Figure 5 nanomaterials-14-01135-f005:**
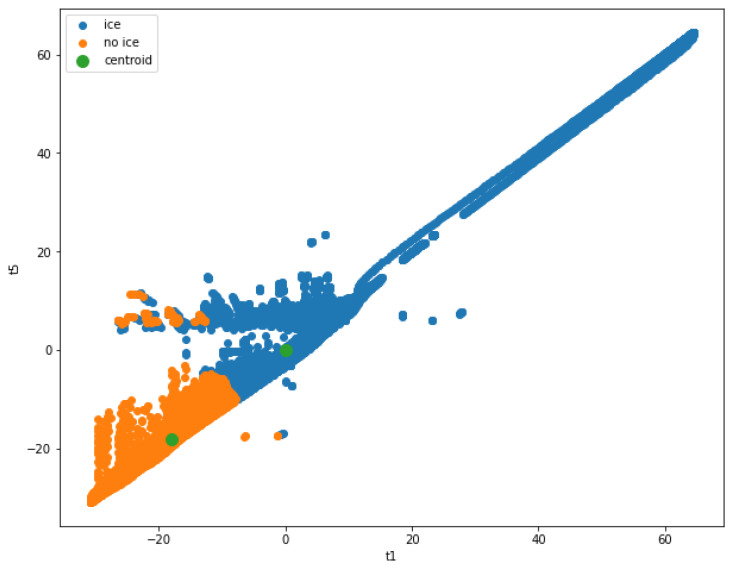
Ice detection using K-Means algorithm.

**Figure 7 nanomaterials-14-01135-f007:**
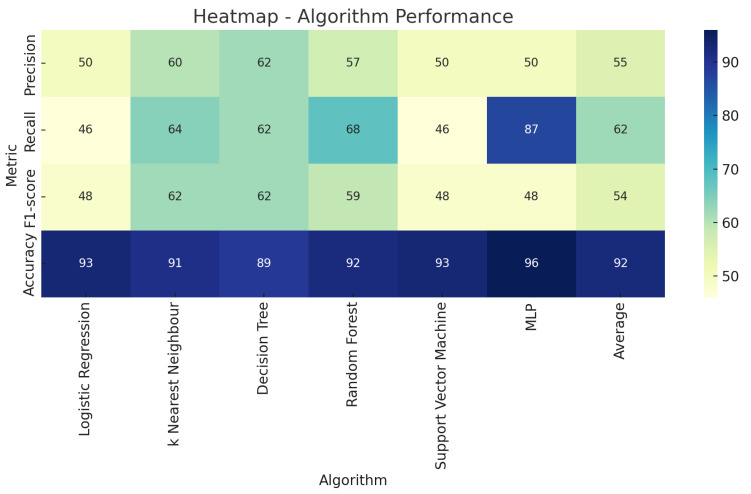
Heatmap comparing the performance of six machine learning algorithms across four metrics.

**Figure 8 nanomaterials-14-01135-f008:**
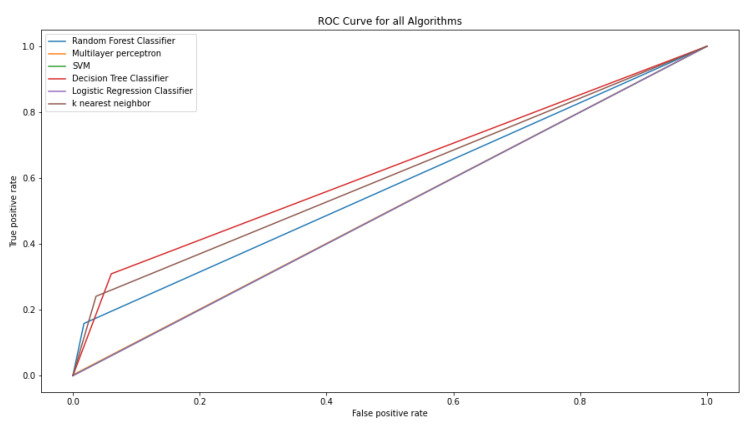
Receiver Operating Characteristic curve for all classifier models.

**Table 1 nanomaterials-14-01135-t001:** Sample data collected using the passive method.

s/n	Time	T_1_ (°C)	T_2_ (°C)
1	18:42:55.632	6.732	6.760
2	18:42:55.797	6.621	6.683
3	18:42:56.009	6.348	6.487
4	18:42:56.216	6.041	6.213
5	18:42:56.430	5.704	5.830
6	18:42:56.590	5.406	5.545
7	18:42:56.798	5.028	5.214
8	18:42:57.007	4.686	4.852
9	18:42:57.213	4.363	4.529
10	18:42:57.420	4.040	4.239

**Table 2 nanomaterials-14-01135-t002:** Values obtained after several experiments.

s/n	Peak Properties	Unit	Passive Mode
1	Maximum Height	°C	2.0
2	Minimum Height	°C	0.8
3	Maximum Width	s	1.5
4	Minimum Width	s	0.5

**Table 3 nanomaterials-14-01135-t003:** Data fitted for the supervised learning algorithm.

t1	t2	t3	t4	t5
6.732	6.621	6.348	6.041	5.704
6.621	6.348	6.041	5.704	5.406
6.348	6.041	5.704	5.406	5.028
6.041	5.704	5.406	5.028	4.686
5.704	5.406	5.028	4.686	4.363

**Table 4 nanomaterials-14-01135-t004:** Logistic parameters for the regression model.

Parameters	Values
t1 Coefficient	−0.0326796
t2 Coefficient	0.02786471
t3 Coefficient	−0.02323838
t4 Coefficient	0.10516002
t5 Coefficient	0.04234012
Intercept	−6.59001252

## Data Availability

All system protocols are publicly accessible and can be found at the Université Libre de Bruxelles.

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
