# Peer review of "Innovative AI-Enhanced Ice Detection System Using Graphene-Based Sensors for Enhanced Aviation Safety and Efficiency"

_nanomaterials, 2024, doi:10.3390/nano14131135_

Round 1
Reviewer 1 Report
Comments and Suggestions for Authors
Authors deliver a quite interesting application of graphene-based thermoresistors for ice detection in aircrafts towards an intelligent actuator system to remove ice for safety reasons. Obviously, the topic is challenging and of high practice interest. The major concern is only its suiting the Nanomaterials’ scope and audience which consists of material science specialists. Here, we don’t see any material (graphene) characterization and functionalization for the task. From this viewpoint, authors are advised to elaborate more properly Introduction in order to consider which materials are used or could be used for the application and why graphene is employed. The text at p.5, lines 179-189, partly meets this target. Conclusions should highlight a benefit, if so, to employ graphene. Otherwise, this paper better suits Sensors journal where it could find a more motivated audience.
Other issues to be resolved:
1) while discussing a structure of the multilayered structure of the sensor system at p. 3, authors are advised to provide a sketch of the structure;
2) while discussing rather detailed the machine learning algorithms, authors do not provide any data on samplings to learn, which conditions they are taken and which variations are accounted for providing possible modifications of the measurands including a pressure (if we talk about aircrafts); in particular, authors note “two datasets” (line 401) which are not discussed in Results;
3) numerous statements/discussions in the text require supporting by literature references; see, for instance, lines 32, 61, 92, 110, 112, 204, 225, 231, 248, 292, 308, 322.
Comments on the Quality of English Language
English is almost fine.
Author Response
Dear Reviewer,
Thank you very much for your valuable feedback and suggestions. We have carefully addressed all your comments and have attached a document with detailed responses and revisions. We appreciate your time and effort in reviewing our manuscript and believe your insights have significantly improved the quality of our work.
Thank you once again for your support and consideration.
Best regards,

Reviewer 2 Report
Comments and Suggestions for Authors
In their manuscript, Farina D. et al. tackle the essential problem of detecting and managing ice formation on aircraft surfaces. The authors propose a promising solution based on graphene-based sensors implemented into the carbon-fiber reinforced polymer employed for aircraft construction, complemented by machine-learning algorithms for data processing to detect ice formation and deal with it through heating.
This manuscript is compelling and well-written, pushing the boundaries of the practical implementation of smart sensors across various fields. The work is done at a decent level of scholarly level, fully realizing the sensor device from both hardware and software perspectives. However, a few omissions require attention before this work can be considered for publication in Nanomaterials in terms of Minor revision, which will further enhance its impact.
1. A review of using graphene-based materials for various smart sensors, including e-nose devices, should be added to Section 2.2 (Lines 178-191). See, for instance, Rabchinskii, M. K. et al. ACS Appl. Mater. Interfaces 2023, 15, 28370–28386 / Carbon 2021, 172, 236−247; Tan, W. C.; Ang, K.-W. Adv. Electron. Mater. 2021, 7, 2001071.
2. Could the authors clarify the choice of Teflon as a mold in their experimental setup? Understanding the rationale behind this decision would provide valuable context for the readers.
3. Page 3, line 125 – “piece of graphene each 0.5 cm” – please specify all the graphene strips' dimensions. Along with the thickness of the graphene layer – was it monolayer or multilayer? Was it free-standing graphene deposited over the prepreg layer or a graphene layer on a supporting substrate (PMMA, etc.)?
4. Figure 2—The X-axis name in Figure 2a is missing. In Figures 2c and 2d, Time and (s) should be separated by space.
5. Section 4, Results. 1. Why are the data points distributed so unevenly in the space surrounding the centroids? Is there any specific reason for such irregular distribution, especially for the ice data points, forming a diagonal line?
6. Page 12, line 363 – any suggestions on which additional features should be added to enhance the precision level?
7. What should be the mean distance between the graphene resistance strips in the designed sensing device to detect ice formation efficiently?
8. The article lacks visual materials demonstrating how the testing with and without ice formation was performed. Introducing such images will improve the quality of the presentation and enhance the readers’ interest.
Author Response

(The authors gave the same response as above.)

Round 2
Reviewer 1 Report
Comments and Suggestions for Authors
Authors have properly updated the manuscript to account for concerns. It could be published out. I wish authors to reach a real application of this graphene-based system to our all benefits